# Mechanical Force-Induced Blue-Shifted and Enhanced Emission for AIEgens

**DOI:** 10.3390/bios12111055

**Published:** 2022-11-21

**Authors:** Chang-Sheng Guo, Xiao-Long Su, Yu-Ting Yin, Bo-Xuan Zhang, Xin-Yi Liu, Rui-Peng Wang, Pu Chen, Hai-Tao Feng, Ben-Zhong Tang

**Affiliations:** 1AIE Research Center, Shaanxi Key Laboratory of Phytochemistry, College of Chemistry and Chemical Engineering, Baoji University of Arts and Sciences, Baoji 721013, China; 2School of Science and Engineering, Shenzhen Institute of Molecular Aggregate Science and Engineering, The Chinese University of Hong Kong, Shenzhen 518172, China

**Keywords:** aggregation-induced emission, mechanochromic luminescence, blue-shift, cyanostilbene, enhanced emission

## Abstract

Mechanochromic (MC) luminescence of organic molecules has been emerging as a promising smart material for optical recording and memory devices. At the same time, pressure-induced blue-shifted and enhanced luminescence are rarely reported now. Herein, a series of cyanostilbene-based AIEgens with different substituents were synthesized to evaluate the influence of morphology transformation and push-pull electronic effect on the MC luminescence. Among these luminophores, compound **1** with one cyano group and diethylamino group was more susceptible to mechanical stimuli and obtained blue-shifted and enhanced fluorescence in response to anisotropic grinding. Powder X-ray diffraction patterns indicated that the MC behaviors were ascribed to the solid-state morphology transition from crystal-to-crystal. Analysis of crystal structures revealed that loose molecular packing is a key factor for high high-contrast MC luminescence. The smart molecular design, together with the excellent performance, verified that luminophores with twisted structures are ideal candidates for MC luminogens.

## 1. Introduction

Stimuli-responsive luminescent materials have received considerable attention in the fields of data storage, information security, and fluorescent probes [1,2,3,4,5]. Various mechanochromic (MC) materials are developed based on organic molecules [6], metal complexes [7], and polymers [8], with controllable luminescence upon mechanical pressing, grinding, and rubbing, which is an important class of smart stimuli-responsive materials. According to previous pieces of literature, MC luminescence mainly results from the altering chemical structure (cleavage and formation of covalent bonds) or solid-state morphology change from crystal-to-amorphous [9], crystal-to-crystal [10], amorphous to crystal [11], or conformational planarization [12]. However, the existing mechanisms still cannot be utilized for all molecular designs. Therefore, developing novel MC materials and exploring the mechanism is beneficial to shed light on the relationship between molecular structure/packing and corresponding MC luminescence to further make a clear picture of MC molecular design.

To date, most reported MC materials afforded a red-shifted and decreased emission in response to anisotropic force or isotropic compression [13,14,15,16,17,18], which is primarily attributed to the conformational planarization under high pressure to generate a low-energy emission species. However, only a few examples of blue-shifted emission have been reported upon mechanical stimuli [19,20,21,22,23,24]. For example, Zou and co-workers reported several carbon dots with blue-shifted luminescence under isotropic pressure [19,20,21]. However, due to no clear structural information, their structure–property relationship is quite difficult to investigate at the molecular level. Moreover, Yang constructed a tetraphenylethylene (TPE)-based luminogen showing a blue-shifted and enhanced emission in view of the cooperative effect of aggregation-induced emission and energy-transfer suppression [23]. Xu found an organic co-crystal presenting blue-shifted fluorescence under anisotropic grinding but a red-shifted emission under isotropic compression [24]. Zhang and co-workers demonstrated a boron diketonate crystal exhibiting a blue-shifted emission under tension but a red-shifted emission under grinding and compressing [25]. Taking these into account, the design of MC material with remarkably blue-shifted and turn-on emission is still a challenge.

Conventional luminophores with planar structures often emitted weak and low-contrast MC luminescence in the solid state because of the aggregation-caused quenching (ACQ) effect, which seriously inhibited their real-world applications [26,27]. Recently, Tang reported a series of propeller-shaped molecules which are non- or weakly emissive in solution but emit strongly in the aggregated state. Such a phenomenon is exactly opposite to the ACQ effect and is termed aggregation-induced emission (AIE) by Tang [28,29,30]. The popular mechanism for AIE is the restriction of intramolecular motion (RIM), including intramolecular rotations and vibrations. By virtue of the distinct photoluminescence (PL) behavior, AIE luminogens (AIEgens) were widely developed to be chem/biosensors and MC materials [31,32,33,34,35,36]. The twisted conformation of AIEgens furnished a loose packing in the crystalline state, which is apt to occur in the transformation between different phases by mechanical stimuli. Until now, many AIE-active MC luminogens have been reported by Chi [37], Weder [38], Tian [24], Li [39], Park [40], and so forth. However, those MC materials based on AIEgens with high-contrast and blue-shifted emission switching are rare.

In this text, a series of cyanostilbene-based AIEgens with different substituents, namely molecules **1**–**6**, were prepared to study the influence of morphology transformation and push-pull electronic effect on the MC luminescence in the solid state. By changing the substituent from the electron-donating diethylamino group to the electron-withdrawing nitro group, the resultant AIEgens afforded remarkable differences in MC luminescence under anisotropic grinding. It is worth noting that molecule **1** showed a turn-on fluorescence accompanied by a blue shift of 28 nm upon grinding. Analysis from powder X-ray diffraction (PXRD) indicated that MC behaviors of these AIEgens were ascribed to the solid-state morphology transition from crystal-to-crystal. These results revealed that luminophores with twisted structures are more susceptible to mechanical stimuli to easily obtain high-contrast MC materials.

## 2. Materials and Methods

**Materials:** All reagents and solvents are chemical pure (CP) grade or analytical reagent (AR) grade and are used as received unless otherwise indicated.

**Measurements:** ^1^H NMR and ^13^C NMR spectra were obtained by an Agilent NMR Systems 400 MHz NMR Spectrometer USA at 298 K (Santa Clara, CA, USA). High-resolution mass spectra (HRMS) were obtained by use of a Bruker Compact TOF mass spectrometer in electrospray ionization mode (ESI+) (Billerica, MA, USA). Absorption spectra were recorded on a YOKU INSTRUMENT TS2023 UV-vis spectrophotometer. Solid absorption spectra were recorded on a Perkin Elmer Lambda 1050+ spectrophotometer (Waltham, MA, USA). Fluorescence spectra were collected on a HORIBA FLOUROMAX-4 fluorophotometer at 298 K (Kyoto, Japan). Powder X-ray diffraction (PXRD) was collected on a Rigaku DMAX U1TlMAIV diffractometer using CuKα irradiation (Tokyo, Japan). The lifetimes were measured on an Edinburgh FLS1000 fluorescence spectrophotometer equipped with a continuous xenon lamp (Xe1) (Edinburgh, UK). The surface morphology of the samples was analyzed using a scanning electron microscope SEM, FEI Quanta FEG 250. Single crystal data were collected on a Bruker Smart APEXII CCD diffractometer using graphite monochromated Mo Kα radiation (λ = 0.71073 Å) or Cu Kα radiation (λ = 1.54184 Å) (Bremen, Germany).

**Synthesis of molecules 1–6:** Molecular engineering played a vital role in materials science and pharmaceutical study. Building up a big molecular library of AIEgens will help to pave the way for unraveling the underlying mechanism of MC luminescence. Cyanostilbene, as a typical AIEgen, has been widely used to construct various luminescent materials in recent years. Here, the cyanostilbene unit was chosen to create a group of MC emitters. The molecules **1**–**6** were facilely synthesized by the Knoevenagel reaction of benzaldehyde and (4-cyanomethyl-phenyl)-acetonitrile. All the AIEgens were characterized by ^1^H NMR, ^13^C NMR, and high-resolution mass (HRMS) spectrometry (Appendix A). Full synthetic details, NMR, and HRMS spectroscopy can be found in the Appendix A.

According to the general procedure (see Appendix A), product **1** was obtained in (0.66 g, yield of 80%). ^1^H NMR (400 MHz, CDCl_3_) δ = 7.85 (d, *J* = 9.2 Hz, 2 H), 7.65–7.62 (m, 2 H), 7.43–7.40 (m, 3 H), 7.33–7.29 (m, 1 H), 6.70 (d, *J* = 9.2 Hz, 2 H), 3.43 (q, *J* = 7.2 Hz, 4 H), 1.22 (t, *J* = 8.2 Hz 6 H). ^13^C NMR (100 MHz, CDCl_3_) δ 149.50, 142.65, 135.85, 131.74, 128.98, 127.93, 125.53, 120.99, 119.72, 111.24, 103.90, 44.64, 12.74. ESI^+^ HRMS *m*/*z* calcd. for C_19_H_21_N_2_ 277.1699 [M+H]^+^, found 277.1697 [M+H]^+^.

According to the general procedure (see Appendix A), product **2** was obtained in (0.74 g, yield of 82%). ^1^H NMR (400 MHz, CDCl_3_) δ 7.87 (d, *J* = 9.2 Hz, 2 H), 7.71 (d, *J* = 8.8 Hz, 2 H), 7.66 (d, *J* = 8.8 Hz, 2 H), 7.46 (s, 1 H), 6.70 (d, *J* = 9.2 Hz, 2 H), 3.45 (q, *J* = 7.2 Hz, 4 H), 1.23 (t, *J* = 7.2 Hz, 6 H). ^13^C NMR (100 MHz, CDCl_3_) δ 150.15, 144.64, 140.28, 132.55, 132.35, 125.55, 120.13, 118.83, 118.63, 111.17, 110.74, 101.22, 44.57, 12.55. ESI^+^ HRMS *m*/*z* calcd. for C_20_H_20_N_3_ 302.1652 [M+H]^+^, found 302.1644 [M+H]^+^.

According to the general procedure (see Appendix A), product **3** was obtained in (0.60 g, yield of 81%). ^1^H NMR (400 MHz, CDCl_3_) δ 7.94–7.91 (m, 2 H), 7.81–7.78 (m, 2 H), 7.76–7.73 (m, 2 H), 7.63 (s, 1 H), 7.51–7.48 (m, 3 H). ^13^C NMR (100 MHz, CDCl_3_) δ 145.03, 138.94, 133.12, 132.97, 131.68, 129.79, 129.31, 126.69, 118.31, 117.28, 112.89, 110.10. ESI^+^ HRMS m/z calcd. for C_16_H_10_N_2_Na 253.0736 [M+Na]^+^, found 253.0736 [M+Na]^+^.

According to the general procedure (see Appendix A), product **4** was obtained in (0.85 g, yield of 92%). ^1^H NMR (400 MHz, CDCl_3_) δ 7.80–7.77 (m, 4 H), 7.76–7.73 (m, 2 H), 7.65–7.61 (m, 2 H), 7.56 (s, 1 H). ^13^C NMR (100 MHz, CDCl_3_) δ 143.50, 138.58, 133.02, 132.62, 131.94, 131.07, 126.70, 126.21, 118.21, 117.03, 113.13, 110.77. ESI^+^ HRMS *m*/*z* calcd. for C_16_H_9_BrN_2_Na 330.9841 [M+Na]^+^, found 330.9841 [M+Na]^+^.

According to the general procedure (see Appendix A), product **5** was obtained in (0.80 g, yield of 95%). ^1^H NMR (400 MHz, CDCl_3_) δ 8.28 (d, *J* = 8.8 Hz, 2 H), 7.94 (d, *J* = 8.8 Hz, 2 H), 7.81 (d, *J* = 8.8 Hz, 2 H), 7.60 (s, 1 H), 7.01 (d, *J* = 8.8 Hz, 2 H), 3.89 (s, 3 H). ^13^C NMR (100 MHz, CDCl_3_) δ 162.56, 147.66, 145.14, 141.23, 132.07, 126.47, 125.83, 124.44, 117.85, 114.80, 106.33, 55.68. ESI^+^ HRMS *m*/*z* calcd. for C_16_H_12_ N_2_O_3_Na 303.0740 [M+Na]^+^, found 303.0739 [M+Na]^+^.

According to the general procedure (see Appendix A), product **6** was obtained in (0.67 g, yield 80%).^1^H NMR (400 MHz, CDCl_3_) δ 8.33 (s, 1H), 8.12–8.10 (dd, *J* = 8.8, 2.0 Hz, 1H), 7.95–7.92 (d, *J* = 9.2 Hz, 2H), 7.89–7.87 (m, 1H), 7.84–7.81 (m, 2H), 7.77 (s, 1H), 7.76–7.73 (M, 2H), 7.62–7.54 (m, 2H). ^13^C NMR (100 MHz, CDCl_3_) δ 144.97, 139.07, 134.69, 133.15, 132.96, 131.57, 130.63, 129.15, 129.09, 128.41, 127.98, 127.23, 126.65, 125.19, 118.34, 117.50, 112.80, 109.87. ESI^+^ HRMS *m*/*z* calcd. for C_20_H_12_N_2_Na 303.0893 [M+Na]^+^, found 303.0889 [M+Na]^+^.

## 3. Results and Discussion

### 3.1. Photophysical Spectra

The photophysical properties of **1**–**6** were investigated in tetrahydrofuran (THF) solution and condensed state. UV-vis spectra of **1**–**6** showed the main absorption peak ranged from 320 to 425 nm due to an intramolecular charge transfer (ICT) transition (Appendix A). PL spectra were studied in THF and THF/water mixture to study their fluorescence behaviors in the aggregated state. Molecules **1** and **2** emitted visibly blue and blue-green fluorescence in THF, respectively (Appendix A). When adding water (poor solvent) in THF, the PL intensity of **1** gradually enhanced with increasing water fraction from 0 to 70%. Further increasing the water fraction to 90%, its emission dramatically quenched at 524 nm with a low PL efficiency, revealing that molecule **1** is not a typical AIEgen. This is consistent with its solid-state emission, which is also weakly emissive (Figure 1). It is speculated that molecule **1** has great potential to serve as a turn-on MC emitter in response to mechanical force. Upon raising the water fraction of **2** to 90%, it always exhibited an upward trend in the PL intensity, indicating an aggregation-induced enhanced emission (AIEE) property. For molecules **3**–**6**, they are almost non-emissive in THF. When the water fraction exceeded 80%, their PL intensities sharply enhanced, showing a typical AIE behavior (Appendix A). The emission wavelength of crystalline powders of **2**–**6** covered from 439 to 562 nm with bright solid-state fluorescence (Figure 1). Then, the PL spectra in polymethyl methacrylate (PMMA) doping film investigated with compound **1** at 1–3% *w*/*w*. Then, 1 g PMMA and 1, 2, 3 wt% **1** were dissolved in 10 mL THF, followed by ultrasonication at room temperature for 2 h. Take the above solution of 60 μL to the quartz glass sheet (1 cm × 1 cm) on the spin coater and spread evenly. Adjusting the rotation speed of 2000 r/min and rotating for 40 s, the quartz glass sheets coated with the sample were used for the film experiments system. The PL intensity of **1** was decreased with increasing the ratio of **1** in PMMA film (Appendix A), implying their molecular interaction strengthened. Such a result is consistent with the aggregated emission of **1**. The thermal behaviors were investigated by thermogravimetric analysis (TGA) under a nitrogen atmosphere. AIEgens **1**–**6** showed a thermal decomposing temperature of 253, 336, 268, 294, 301, and 329 °C, respectively, demonstrating high thermal stability (Appendix A).

### 3.2. MC Behaviors of Molecules **1–6** under Grinding

Pristine emitters **1–6** were ground by using a pestle in a mortar. Then the powders were collected for measurement at room temperature. The crystalline powders of **1**–**6** were recrystallized in dichloromethane. Under 365 nm UV illuminations, molecule **1** is weakly emissive at 524 nm. After grinding, **1** blue-shifted to 496 nm with an intense blue-green fluorescence (Figure 2A and Appendix A). The absolute fluorescence quantum yields (Φ_F_) of **1** were increased from 0.1+/−0.02% to 4.3+/−0.02% after grinding, indicating a turn-on blue-shifted MC material was successfully developed. The lifetime of **1** had no obvious change (τ = 0.43 ns) after grinding (Appendix A). When another cyano group was introduced to the left phenyl ring in the structure, the obtained molecule **2** had no wavelength change under grinding, while the emission efficiency and lifetime showed an obvious decrease (Figure 2B and Appendix A). In contrast to **2**, the diethylamino group at the para-position of the right phenyl ring was replaced with hydrogen and bromine to give molecules **3** and **4**, respectively. However, the as-prepared crystalline powders were MC silence under grinding (Figure 2C,D). Then the electron-withdrawing nitro group was modified in molecule **5**. It also showed no response to mechanical grinding (Figure 2E). Upon replacing the right phenyl ring with the naphthyl ring, surprisingly, the obtained molecule **6** showed a red-shift by 10 nm from 493 to 503 nm (Figure 2F). Meanwhile, the fluorescence QY enhanced from 5.4+/−0.02% to 6.3+/−0.02% after grinding. For molecules **3–6**, the lifetimes of ground powders exhibited a little decrease compared with their pristine solids (Appendix A). Solid-state UV-vis spectra of **1**–**6** were also studied by grinding the crystalline powders. Then we investigated the luminescence spectra of crystals **1**–**6** under the different excitation wavelengths. The results showed that the excitation wavelength has no effect on the luminescence spectra (Appendix A). As shown in Appendix A, it was found that all materials had a minor change in their absorption spectra in response to grinding. The reversibility of MC luminescence of **1**–**6** was investigated by fuming organic solvents. As shown in Figure 2A, the wavelength of the fumed sample **1** cannot restore to its originated state, but the PL intensity recovered, implying it may occur morphology change. Except for molecule **3**, the other AIEgens showed negligible wavelength change after fuming.

Then, powder X-ray diffraction (PXRD) measurements were carried out to obtain insight into their MC properties. As shown in Figure 3, the pristine crystals of **1**–**6** showed many sharp and intense diffraction peaks, suggestive of their good crystallinity and well-ordered molecular packing. After grinding, the PXRD patterns of **1**, **2**, **3**, **5**, and **6** gave rise to some new diffraction peaks with intense intensity, indicating that their molecular arrangements have occurred some changes. Molecule **4** is MC silent and found there was no change in the PXRD pattern upon heavy grinding. The above results revealed that mechanical grinding could not transform the crystalline state into the amorphous state. Therefore, the MC behaviors for **1** may be attributed to the change in the solid-state crystal-to-crystal morphology transition. After fuming, the recrystallized **1** exhibited a different PXRD pattern in contrast to its pristine and ground samples. Thereby, it is anticipated that the mechanical force changed the nano-assemblies to another packing mode. That’s why the fumed sample **1** cannot restore to its originated state. From these PXRD results, we can see that MC characteristic is indeed associated with the alterations of packing modes and crystalline state.

Frontier molecular orbitals were calculated at the B3LYP/6–311G* level by using density functional theory (DFT) to study their optical properties. As shown in Figure 4, although the DFT calculation showed compound **1** had no very clear separation of the highest occupied molecular orbital (HOMO) and the lowest unoccupied molecular orbital (LUMO), after careful analysis, we can see that the distribution of HOMO and LUMO is non-uniform in compound **1**. The electrons of HOMO are mainly located on the right segment of **1**, especially on the electron-donating diethylamino group. While the electrons in LUMO mainly resided on the left part. The electron cloud density of the phenyl ring in the box obviously enhanced, and the electrons on the diethylamino group decreased largely in LUMO. Thus, we think it existed a weak ICT transfer in compound **1**. The energy gap between the LUMO and the HOMO was calculated to be 3.585 eV. Molecules **2** and **5** have dipolar architectures due to the electron-withdrawing ability of cyano and nitro groups. The HOMO mainly centered on the electron-donating groups, but the LUMO lay on the cyanostilbene unit. The energy gap of **2** and **5** was 3.228 and 3.428 eV, respectively. For molecules **3**, **4,** and **6**, the HOMO and LUMO resided on the whole molecule, and the energy gap was estimated to be 3.897, 3.764, and 3.610 eV, respectively. Among these AIEgens, **2** and **5** possessed a much smaller energy gap. Theoretically, they should have much redder emission. Actually, **2** and **5** emitted yellow fluorescence in powder, and the other molecules exhibited blue or green emission, which was in accordance with the calculation (Figure 1).

In order to obtain insight into the underlying origins of the remarkable blue-shift and turn-on MC luminescence, the intermolecular interactions in the single-crystal state were studied (Figure 5). Two single crystals suitable for X-ray diffraction were obtained by slow evaporation of the solution of **1** and **2** in mixed CHCl_3_/CH_3_OH solvents. As illustrated in Figure 5A, **1** adopted a twisted molecular conformation in the crystalline state. The dihedral angle between the left phenyl ring and the plane of the double bond is 45.45°. Such a distorted molecular structure can self-assemble into a loose packing (Figure 5C), which might readily slide upon mechanical grinding to result in MC luminescence. Moreover, some C-H···π (*d*_CH···π_ = 2.896 Å) interactions were obtained in the neighboring molecules (Figure 5B). While the crystal structure of **2** showed a planar molecular conformation and the adjacent molecules arranged in a head-to-tail manner (Figure 5D). Some short contacts, such as C-H···π (*d*_CH···π_ = 3.571 Å), C≡N···H-C (*d*_CN···HC_ = 3.373, 3.104, 2.621, 2.274 Å), π–π (*d*_π-π_ = 3.573 Å), C-H···C-H (*d*_CH···HC_ = 3.251 Å) interactions, were observed in adjacent molecules (Figure 5E). Moreover, these molecules assembled into ladder-like nanostructures with a uniform arrangement (Figure 5F). This packing mode of **2** in its pristine state is beneficial for solid-state emission.

Their microstructure change was also confirmed by scanning electron microscopy (SEM) (Figure 6A–L). As shown in Figure 6A, molecule **1** afforded a relatively loose surface in pristine form. Upon grinding, sample **1** gave cotton-like microstructures. As a comparison, molecules **2**–**6** self-assembled into many rod-like or belt-shaped nanostructures in their pristine state. After mechanical grinding, many unordered structures were generated. It is worth noting that the PXRD results indicated these ground samples still exhibited intense diffraction peaks. These results revealed that the molecular stacking manners had been changed after the external environmental stimulation, which played an important role in their MC luminescence.

## 4. Conclusions

In summary, a series of cyanostilbene-based AIEgens were designed and synthesized to investigate their mechanochromic properties through molecular engineering. Among these luminophores, compound **1** with one cyano group and diethylamino group afforded a turn-on and blue-shifted emission in response to anisotropic grinding. Then photophysical test, PXRD patterns, theoretical calculations, and analysis of crystal structural information suggested that the change of molecular packing mode and morphology transition from crystal-to-crystal are responsible for the blue-shifted emission after grinding. All these results showed that this strategy would be promising for designing excellent mechanochromic materials based on AIEgens.

## Figures and Tables

**Figure 1 biosensors-12-01055-f001:**
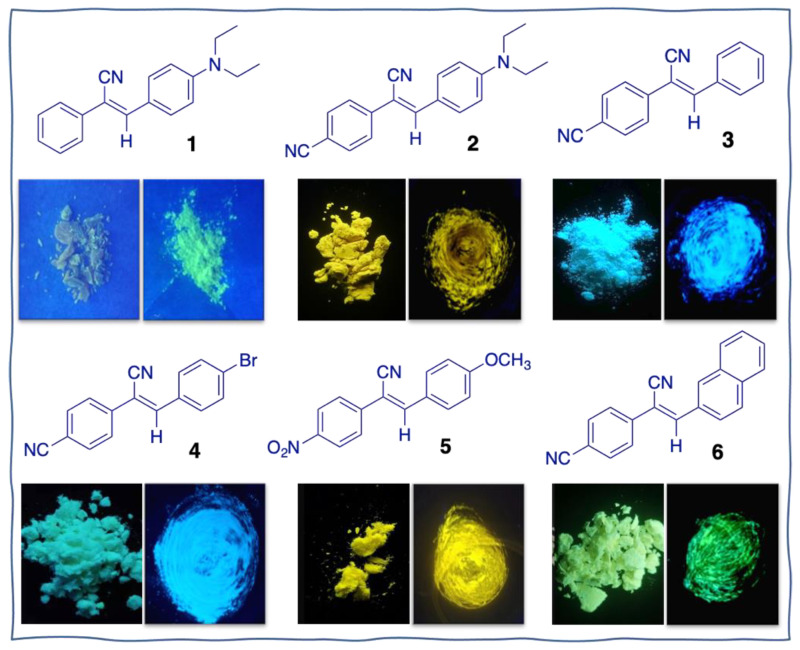
Molecular structures of **1**–**6** and their photographs were taken before and after grinding under 365 nm UV light.

**Figure 2 biosensors-12-01055-f002:**
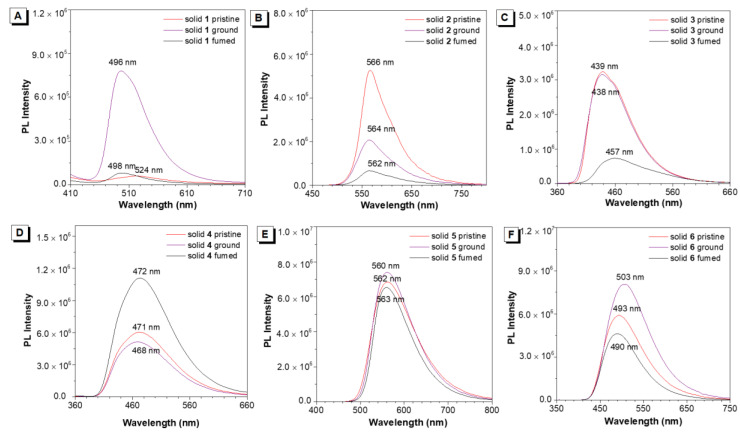
PL spectra of molecules **1**–**6** as pristine and ground solids. Molecule **1** (Excitation wavelength = 389 nm), molecule **2** (Excitation wavelength = 425 nm), molecule **3** (Excitation wavelength = 320 nm), molecule **4** (Excitation wavelength = 326 nm), molecule **5** (Excitation wavelength = 365 nm), molecule **6** (Excitation wavelength = 334 nm).

**Figure 3 biosensors-12-01055-f003:**
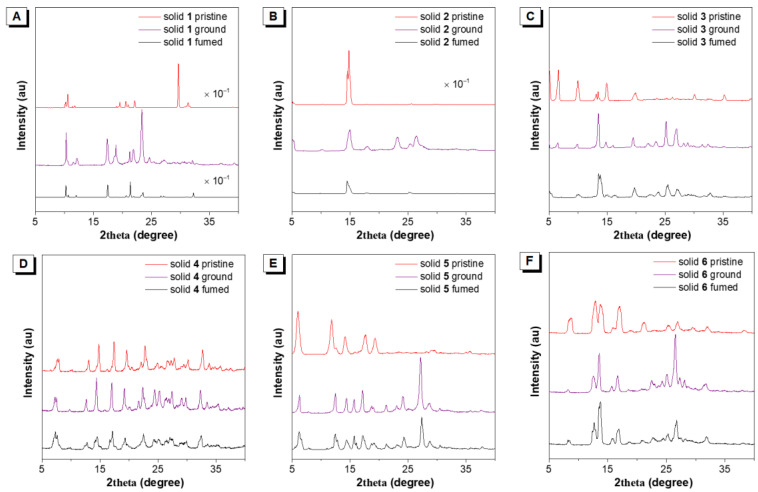
(**A**–**F**) The PXRD patterns of molecules **1**–**6** as the pristine, ground, and fumed solids.

**Figure 4 biosensors-12-01055-f004:**
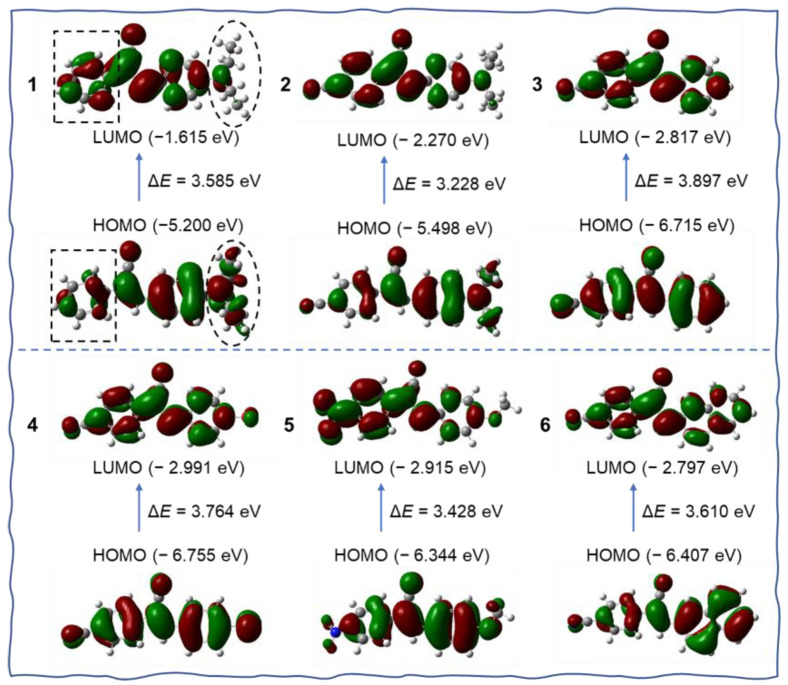
The molecular orbitals of molecules **1**–**6** are calculated by the DFT method.

**Figure 5 biosensors-12-01055-f005:**
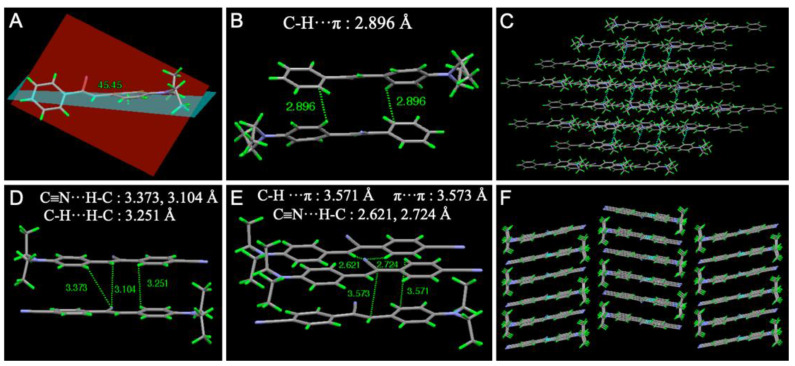
The crystal structures and intermolecular interactions of (**A**–**C**) **1** and (**D**,**F**) **2**. Hydrogen: Green, Carbon: Grey, Nitrogen: Purple.

**Figure 6 biosensors-12-01055-f006:**
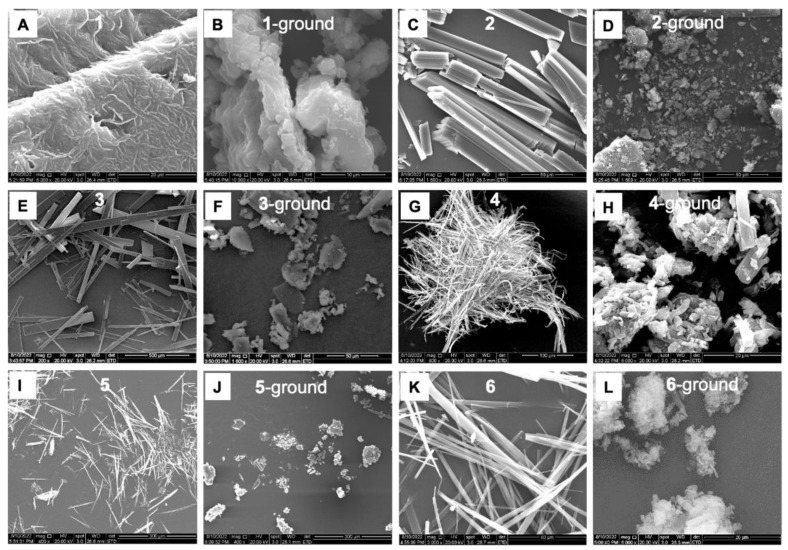
SEM images of **1**–**6** were taken before and after grinding.

## Data Availability

Not applicable.

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
