# Peer review of "Mechanical Force-Induced Blue-Shifted and Enhanced Emission for AIEgens"

_biosensors, 2022, doi:10.3390/bios12111055_

Round 1
Reviewer 1 Report
the manuscript titled “Mechanical Force-Induced Blue-Shifted and Enhanced Emission for AIEgens”, the authors report the synthesis and physicochemical properties of a series of cyanostilbene derivatives. The study of mechanochromic properties for AIEgens based on cyanostilbene fragments is commonly reported, and the redaction of the manuscript does not show an important scientific contribution. In this way, in my consideration, this version of the manuscript is unacceptable to be accepted for publication.
Please, authors should include a complete description of synthesis for derivatives 2, 3, 4, 5, and 6. On page 3, line 115, the authors indicate “The synthesis route is similar to molecule 1” however, in the synthesis of 2 are employed other starting materials. Recommendation: describe a general methodology.
Authors should include the error in the experiments of high-resolution mass spectrometry (page 3, lines 114, 121, 127, 133, 139, and 146)
On page 4, line 151. Authors should use the correct description of electronic transition π®π*
Please, the authors should improve the redaction of the results and discussion section (on page 4, line 152). To be more clear, authors have to use correctly the term AIE or aggregation enhanced emission (AEE) [DOI 10.1002/agt2.7], because line 153, is described that all fluorophores exhibit emission in THF solution.
Authors should indicate the error in the fluorescent quantum yield to show if the increment in the value is real.
On page 6, lines 215-218. The authors describe the unsymmetrical distribution of electronic density, however, figure 4 is not shown. Also on the same page, line 221 authors indicate the presence of D-A electronic architecture, but only compounds 2 (CN-π-NEt2) and 5 (NO2-π-NEt2) have a dipolar architecture. Discussion of theoretical estimations and conclusions should be improved.
On page 8, line 250. The author should improve the redaction. Because indicate changes in the nanostructure of solids, but the SEM micrographs (Figure 6) are in the scale of micrometers.
Reviewer 2 Report
In this manuscript, photophysical properties of a series of cyanostilbene based AIEgens showing luminescent mechanochromism were described. The authors found that some of reporting compound showed blue shift of the luminescence maximum and enhanced luminescence intensity by grinding the as-prepared crystals. The structure-property relationship and mechanism of such mechanical response of luminescence properties are fairly discussed in the manuscript. I feel that research is well designed, and that reporting materials and phenomena are interesting for many readers of the journal. Thus, the work can be considered for publication. Specific comments are as follows:
Figure 2 demonstrates the blue shift of the luminescence maximum and enhanced luminescence intensity by grinding the as-prepared crystals. However, the shift of the luminescent maximum cannot be recognized well by this figure. The normalized spectra should also be shown for clear understanding the blue shift; those can be added in the supporting materials.
The excitation wavelength should be described in the caption of Figure 2. Are there any dependencies of the excitation wavelength on the luminescence spectra in crystals?
Reviewer 3 Report
Tang and coworkers have synthesized six cyanostilbene-based AIEgens with different electron-deficient/-rich substituents. Their photophysical behaviors in the solution and crystal state have been detailedly investigated by fluorescence, UV-Vis absorption spectroscopy, and theoretical calculations. The diethylamino-substituted AIEgen exhibited obviously blue-shifted and enhanced luminescence on account of the loose molecular packing in the crystal upon grinding. This research provides some references for designing smart luminescent materials that respond to mechanical force. The manuscript has been well organized with clear logic and presentation. I recommend publishing this work in Biosensors after the following comment have been considered.
(1) The tendency (5<6<1) of bandgaps for AIEgens 1, 5, and 6 based on DFT calculation has conflicted with that (1<5<6) observed in their UV-Vis absorption spectra. Please provide a reasonable explanation.
Round 2
Reviewer 1 Report
The new version of the manuscript titled Mechanical Force-Induced Blue-Shifted and Enhanced Emission for AIEgens" shows a considerable improvement. However, some experimental information should be included and expand the discussion concerning the supramolecular arrangement and the PL properties.
Please, the authors should review the assignment of D-A electronic molecular architecture because compound 1 doesn't have this type of architecture. Inclusive, the authors based their conclusions on this observation.
I recommend that authors include the PL for compound 1 in a thin film. Discussion of PL spectra for a thin film of optical inert polymer matrix doped with compound 1 at 1-3% w/w should be included to investigate the effect of intermolecular interactions in emission.
on page 7, line 247. In the line "This packing mode is beneficial for the solid-state emission" the authors should explain what solid of the investigated (pristine, ground or fumed) presents this supramolecular arrangement. Because for pristine solids the PL characteristic is reduced (figure 3A).
Author Response
"Please see the attachment."
